# Cross-Talk between p53 and Wnt Signaling in Cancer

**DOI:** 10.3390/biom12030453

**Published:** 2022-03-15

**Authors:** Qiyun Xiao, Johannes Werner, Nachiyappan Venkatachalam, Kim E. Boonekamp, Matthias P. Ebert, Tianzuo Zhan

**Affiliations:** 1Department of Medicine II, Mannheim University Hospital, Medical Faculty Mannheim, Heidelberg University, Theodor-Kutzer-Ufer 1-3, D-68167 Mannheim, Germany; qiyun.xiao@medma.uni-heidelberg.de (Q.X.); nachiyappan.venkatachalam@medma.uni-heidelberg.de (N.V.); matthias.ebert@umm.de (M.P.E.); 2Division Signaling and Functional Genomics, German Cancer Research Center (DKFZ), and Department Cell and Molecular Biology, Faculty of Medicine Mannheim, Heidelberg University, D-69120 Heidelberg, Germany; johannes.werner@dkfz-heidelberg.de (J.W.); k.boonekamp@dkfz-heidelberg.de (K.E.B.); 3Mannheim Cancer Center, Medical Faculty Mannheim, Heidelberg University, Theodor-Kutzer-Ufer 1-3, D-68167 Mannheim, Germany; 4DKFZ-Hector Cancer Institute at the University Medical Center Mannheim, Theodor-Kutzer-Ufer 1-3, D-68167 Mannheim, Germany

**Keywords:** Wnt, p53, APC, beta-catenin, drug resistance, cancer, metastasis

## Abstract

Targeting cancer hallmarks is a cardinal strategy to improve antineoplastic treatment. However, cross-talk between signaling pathways and key oncogenic processes frequently convey resistance to targeted therapies. The p53 and Wnt pathway play vital roles for the biology of many tumors, as they are critically involved in cancer onset and progression. Over recent decades, a high level of interaction between the two pathways has been revealed. Here, we provide a comprehensive overview of molecular interactions between the p53 and Wnt pathway discovered in cancer, including complex feedback loops and reciprocal transactivation. The mutational landscape of genes associated with p53 and Wnt signaling is described, including mutual exclusive and co-occurring genetic alterations. Finally, we summarize the functional consequences of this cross-talk for cancer phenotypes, such as invasiveness, metastasis or drug resistance, and discuss potential strategies to pharmacologically target the p53-Wnt interaction.

## 1. Introduction

Cancer is the second most common cause for premature death worldwide [1] and, therefore, a major medical challenge. In recent decades, considerable insights into the biology of cancer have been obtained, and this knowledge has been partly translated into novel therapeutic approaches exploiting tumor-specific cellular and molecular vulnerabilities [2,3]. Among the identified hallmarks of cancer, resistance to cell death, mutational and genomic instability, and sustained oncogenic signaling play exceptional roles as druggable targets [4]. The importance of these hallmarks for cancer biology is further highlighted by the high frequency of mutational events observed in genes associated with these key oncogenic processes, including *TP53*, *APC*, *PIK3CA* or *KRAS* [5]. Furthermore, many hereditary cancer syndromes can be traced back to the dysregulation of either mutational stability (e.g., Li-Fraumeni syndrome, Lynch syndrome) or intracellular signaling (e.g., Familial adenomatous polyposis [FAP] syndrome, Cowden syndrome) [6]. Targeting these hallmarks is seen as a fundamental approach to improve cancer therapy. However, while many small molecules directed against these altered processes, such as kinase inhibitors, have been developed and tested in preclinical models, only few compounds were successfully introduced into clinical practice. A potential reason for this failure is that cancer hallmarks are often tightly intertwined in feedback and -forward loops, resulting in molecular and cellular networks that are robust to external perturbations, e.g., drug treatment [7,8]. Hence, deciphering critical intersection points between cancer hallmarks can reveal novel and potent therapeutic targets. Two pathways that play major roles for tumor biology are the p53 and Wnt signaling pathway, as they act as central nodes of many cancer hallmarks. In this review, we summarize how p53 and Wnt signaling interact in cancer through multiple regulatory routes and on different molecular levels [9,10,11]. The mutational landscape of p53 and Wnt associated genes across different cancer entities is outlined, with a focus on co-occurring or mutual exclusive mutational events. The phenotypic consequences of the cross-talk between p53 and Wnt pathways for major tumor phenotypes are highlighted and therapeutic opportunities by targeting this interaction are discussed.

## 2. p53 and Its Role in Cancer

The transcription factor p53, encoded by the *TP53* gene, is one of the most well-studied human genes [12]. p53 functions as a sensor for a variety of cellular stress signals, including DNA damage, oncogene activation and hypoxia. As a response to these stimuli, p53 can trigger apoptosis, cell cycle arrest and senescence through the transcription activation of key regulators (Figure 1A) [6,13]. The critical role of p53 for tumor suppression is well described by loss-of-function mouse models that spontaneously develop tumors in a variety of tissues, such as soft tissue sarcoma or lymphoma [14,15]. Functionally, mutations in *TP53* can lead to either inactivation or gain-of-function of the protein [16,17]. Murine tumor models indicate that tumors arising from either gain- or loss-of-function mutations have distinct phenotypes, underlining that mutant p53 can have oncogenic properties as well [18,19]. A major characteristic of mutant p53 is that it can exert a dominant-negative inhibition of its wild-type counterpart [20,21]. In addition, specific missense p53 mutants are able to actively reshape the interactome of p53, thereby modulating cellular pathways and support cancer proliferation, migration and metastasis [10]. Over recent decades, more diverse cellular effects of p53 were revealed, which contribute to tumor growth control as well. For instance, p53 can regulate cellular metabolism [22] and sensitize cells to ferroptosis-associated cell death [23]. In addition, p53 also controls autophagy [24], cellular differentiation and stem cell renewal [25], which are critical processes for the maintenance of cancers. The functional relevance of these additional mechanisms for tumor suppression is highlighted by several mouse models with specific p53 mutations, which show that p53 retains tumor suppressive activity despite impaired effects on cell cycle and apoptosis regulation [26,27].

### p53 Associated Proteins and Transcriptional Targets

p53 exerts its tumor suppressive function through interaction with a variety of upstream and downstream interaction partners [28]. The main regulator of p53 is the E3 ubiquitin ligase MDM2 [29], which can activate the degradation of p53 by the ubiquitin system [30,31]. The transcription of MDM2 itself can be activated by p53, resulting in a negative feedback loop that balances levels of MDM2 and p53 [32]. Besides ubiquitination, p53 can also be phosphorylated at Ser15, Thr18 or Ser20 residues, which interrupts its interaction with MDM2 [33,34,35]. In untransformed cells, these three residues are not phosphorylated, and p53 is maintained at low levels by MDM2 [34,36]. In the presence of specific stress signals, protein kinases, such as ATM, DNA-PK, CHK1 and CHK2, are activated and phosphorylate p53 at either of the three aforementioned residues, leading to the stabilization of p53 (Figure 1A) [37]. After the stress stimulus is reverted, the these kinases are no longer active, and p53 will be quickly dephosphorylated by phosphatases such as PP1, leading to its subsequent degradation by the accumulated MDM2 [38]. Furthermore, p53 is subject to many other post-translational modifications, such as acetylation and methylation, which can fine-tune its function [39,40]. In concert with its family members p63 and p73, p53 controls the expression of a large network of target genes [41,42]. The main transcriptional targets can be summarized into groups that regulate the cell cycle (p21, Gadd45 and 14-3-3), DNA repair and damage prevention (p53R2, p48 and sestrins), and apoptosis (Bax, Apaf-1, PUMA and NoxA) [41]. In addition, several genes involved in key metabolic processes are transcriptionally controlled by p53 [22]. For instance, p53 regulates glucose metabolism through the expression of GLUT transporters [43] and TIGAR [44], a protein that blocks glycolysis. Furthermore, fatty acid metabolism is also under the transcriptional control of p53 via the expression of AMPKα2, LKB1, and SIRT1 [45]. Finally, transcription-independent effects of p53 were also described, for instance on apoptosis through physical interaction with Bcl-2 family members, providing an additional layer of complexity to the biological function of p53 [46].

## 3. Wnt Signaling

Wnt signaling comprises several evolutionarily highly conserved pathways which are modulated by the family of Wnt proteins (WNTs), and is diverted into canonical and non-canonical Wnt signaling [11]. The Wnt pathway is implicated in crucial biological processes in animals, including embryological development, morphogenesis and tissue organization [9]. WNTs are secreted from cells to exert their biological function in an autocrine or paracrine fashion [47]. The effect of WNTs is mediated by a multitude of receptors and co-receptors including Frizzled (FZDs), LRP or ROR/RYK [48,49]. Canonical Wnt signaling depends on β-catenin (encoded by the *CTNNB1* gene), which transduces Wnt signals into a transcriptional response in cooperation with TCF/LEF transcription factors [50]. The abundance and subcellular distribution of β-catenin is tightly regulated by the destruction complex, a multiprotein complex consisting of its core components APC, GSK3β, AXIN1/2, CKIα and the ubiquitin ligase β-TrCP [28]. By means of phosphorylation and subsequent ubiquitination, the destruction complex primes β-catenin to proteasomal degradation (Figure 1B) [51]. Due to its important biological function, the canonical Wnt signaling is tightly regulated on several levels. On the receptor level, the abundance of WNTs, their respective antagonists (SFRPs, WIFs, DKKs, NOTUM) and their cognate membrane receptors influence canonical Wnt signaling [48,52]. Furthermore, R-spondin ligands were found to positively affect Wnt signaling [53]. These secreted proteins bind to the plasma membrane receptors LGR4-6, which in turn inhibit the function of the two E3 ubiquitin ligases *ZNRF3* and *RNF43* [54]. The *ZNRF3*/*RNF43* module targets FZD receptors for lysosomal degradation, which limits its availability at the plasma membrane [55]. Further downstream, the pathway can be regulated by the composition of the destruction complex, and by the interaction and stability of its core components [51]. The destruction complex frequently integrates signals from other oncogenic pathways that modulate Wnt signaling, such as the Ras and Hippo pathways [56,57]. Lastly, transcriptional co-regulators of LEF/TCF transcription factors, such as CBP/p300 or BCL9, can impact the biological effects of the canonical Wnt pathway [9]. In adult human tissues, canonical Wnt signaling is required for the maintenance of stem cells and tissue renewal [58]. Spatiotemporal regulation of Wnt signaling therefore plays a vital role for tissue homeostasis and renewal, which is well described for the intestinal epithelium [59,60,61]. In contrast to canonical Wnt signaling, non-canonical Wnt signaling acts independently of β-catenin. It is initiated by a subgroup of Wnt ligands (e.g., WNT5A and WNT11) in interactions with specific FZDs and ROR/RYK receptors [62]. Non-canonical Wnt signaling cascades include the Wnt planar cell polarity (Wnt-PCP) and Wnt-Ca^2+^ pathway [11]. These pathways modulate processes such as cell migration and polarity and are implicated not only in development and tissue homeostasis, but also in pathophysiological processes such as metastasis [63,64,65]. Due to the key role of Wnt signaling in cell proliferation, polarity and migration, it is not surprising that aberrant Wnt signaling is a vital component of several diseases [9]. In mice, germline deletion of Wnt pathway components is often lethal or leads to severe developmental defects [66]. Upregulation of canonical Wnt signaling occurs in many cancer types, such as colorectal or breast cancer [67]. For further reading on the role of Wnt signaling in cancer, we refer to more comprehensive reviews [9,11,48,68,69,70].

## 4. The Mutational Landscape of p53 and Wnt Pathway Genes in Cancer

Wnt signaling is frequently dysregulated in human cancer. Depending on the type of cancer, the Wnt pathway can be upregulated by different mechanisms, including the overexpression of specific components [71,72], epigenetic dysregulation (e.g., *APC* promoter hypermethylation [73]), somatic mutations [11] or gene fusions [74]. As regulation of Wnt signaling is highly complex, cancer-relevant mutations occur on different levels of the pathway, with specific functional implications. For example, truncating mutations of *RNF43* or *ZNRF3* can result in a ligand-dependent hyperactivation of Wnt signaling [75]. Furthermore, genetic alterations are frequently observed in components of the destruction complex, such as *APC* or *AXIN1* [76,77,78], but can also occur on the level of nuclear signal transducers (*CTNNB1*, *TCF7L2*) [79,80]. These different mutations show a heterogenous distribution across cancer types with some genetic alterations occurring predominantly in selected cancer entities, indicating tissue-specific dependence on differential Wnt pathway activation to enable carcinogenesis [80,81] (Figure 2A). For instance, mutations of *APC* occur as an early driving event in colorectal carcinogenesis and can be observed in ~50% of sporadic colorectal cancers [82,83]. *APC* mutations also occur in 20–25% of small intestinal cancers as well as in 12–32% of gastric cancers [84,85,86]. Heterozygous germline mutations of *APC* predispose to colorectal adenomas and early-age colorectal cancer, which are described by the familial adenomatous polyposis (FAP) syndrome [87]. Mutations of the Wnt pathway are found in several cancer entities besides gastrointestinal cancers. In hepatocellular carcinoma, gain-of-function mutations of *CTNNB1* (20–25%) and loss-of-function mutations of *AXIN1* and *AXIN2* are frequently observed [5]. Furthermore, in endometrial cancer, *RNF43* and *CTNNB1* mutations are regularly detected, with *CTNNB1* mutations being associated with aggressive cancers and poor survival [75,88]. Moreover, different Wnt pathway mutations are implicated in prostate cancer, and a significant number of castration-resistant tumors carry mutations of *CTNNB1*, *APC* or *RNF43/ZNFR3* [89].

*TP53* is the most frequently mutated gene in cancer [5,81]. The Li–Fraumeni-syndrome, a hereditary disposition to multiple types of cancer, is caused by heterozygous germline mutations of *TP53*, indicating a conserved role for tumor suppression across different tissues [90]. Data from the International Agency for Research on Cancer database show that of all observed *TP53* single nucleotide substitutions in cancer, 88% are missense mutations which occur in the DNA-binding domain [91,92]. Known mostly as a tumor suppressor gene, gain-of-function phenotypes of mutant p53 have also been described [91,93]. For instance, missense mutations, but not the deletion of *TP53*, could drive tumor cell invasion and activate Wnt/β-catenin signaling to enhance cancer invasiveness in a colorectal cancer mouse model [94]. In contrast to p53, few mutations are observed for its most important binding partner, MDM2. However, MDM2 amplifications are frequently found in sarcomas and preferentially occur in *TP53* wild-type tumors [58].

An indicator of functional interaction between genes in cancer is mutual exclusion or co-occurrence of genetic alterations [95]. In a study of prostate cancer tissue, mutations in the Wnt/β-catenin signaling pathway were found to occur less frequently in *TP53* mutant than in *TP53* wild type samples [96]. Moreover, *CTNNB1* mutations occurred mutually exclusive to *TP53* mutations in a panel of hepatocellular cancer tissue samples, suggesting that either of the two mutations is sufficient to initiate carcinogenesis [97]. A pan-cancer study revealed the mutual exclusivity of *TP53* mutations with certain driver mutations (such as *VHL* mutations in renal clear cell carcinoma), but not with *APC* mutations in colorectal cancer [81] (see Figure 2B for examples). This corroborates the assumptions of the adenoma-carcinoma-sequence model, which suggests that colorectal cancer requires mutations in both *APC* and *TP53* for progression [81,83]. Taken together, current cancer genomics data indicate a cancer type-specific interplay between mutations in the Wnt and p53 pathway. However, the functional implications of these distinct mutational patterns are not yet understood.

As mutations of the Wnt and p53 pathways play an important role for cancer biology, many studies correlated the presence of these mutations with cancer aggressiveness and patient survival. Occurrence of *TP53* mutations was generally found to be associated with poor survival in cancer patients. In several tumor entities, including hepatocellular carcinoma [98], lung adenocarcinoma [99], CRC [100,101], bladder cancer [102] and breast cancer [103,104], *TP53* mutations predicted poor survival. Furthermore, in a meta-analysis, a correlation between *TP53* mutation and risk of distant metastasis was observed [105]. In contrast, the prognostic value of mutations in components of the Wnt pathway is more diverse. In CRC, studies have shown either an association of *APC* mutations with favorable prognosis or no influence on patient survival [106,107,108]. However, CRC with concomitant *APC*, *KRAS* and *TP53* mutations were found to have a poor prognosis [107], underlining the importance of combined *APC* and *TP53* mutations for the biology of CRC. In other cancer types, *APC* mutations are associated with either reduced or increased survival [109,110]. Similarly, *CTNNB1* mutations are differentially associated with survival. In hepatocellular and lung adenocarcinoma, no association of the *CTNNB1* mutational status with prognosis was found, while in certain subtypes of early stage endometrial carcinoma, *CTNNB1* mutations predicted disease recurrence and reduced disease-free survival [111,112,113]. However, these results need to be interpreted with care, as sample sizes and definition criteria of mutations vary significantly between studies.

## 5. Molecular Interactions between p53 and Wnt Pathway in Cancer

Observations from a variety of tumor models suggest that the p53 and Wnt signaling pathways cooperate with each other to drive tumor initiation and progression [17,80]. Therefore, many studies have focused on identifying molecular mechanisms that underlie this interaction. In the following chapter, we comprehensively summarize the different levels of cross-talk between the p53 and Wnt pathways which were observed in various cancer models (Figure 3).

### 5.1. Interactions of p53 with the β-Catenin Destruction Complex

The autoregulatory loop of β-catenin and p53 is one of the earliest discovered and most well described molecular interactions between both pathways in cancer. In 1999, Damalas et al. demonstrated that the overexpression of β-catenin induced the accumulation of transcriptionally active p53, which triggered an inhibitory p53 response in lung adenocarcinoma cells [115]. This β-catenin effect was caused by the inhibition of the MDM2-mediated proteolytic degradation of p53. The same stabilization of p53 could be triggered by the overexpression of an upstream Wnt pathway component, Dishevelled (DVL) [115]. Later, the same group showed that excess β-catenin caused the constitutive accumulation of p53 via transcriptional upregulation of an alternative reading frame product of the INK4A tumor suppressor locus (P14ARF/CDKN2A), which binds to and inhibits MDM2 [116] (Figure 3A). As a consequence, cellular senescence is induced, which is considered as a safeguard mechanism against Wnt-induced carcinogenesis [116]. Correspondingly, in p53-impaired conditions, either caused by the expression of a p53 loss-of-function variant or ablation of P14ARF, β-catenin regains its oncogenic effects and is able to drive carcinogenesis, most notably in colorectal cancer [116,117].

Interestingly, high levels of wild-type, active p53 were found to downregulate β-catenin, thus generating a bidirectional negative feedback loop between p53 and Wnt signaling [118]. The underlying mechanism relies on ubiquitination and proteasomal degradation of β-catenin, which requires the function of GSK3β [118], a core component of the destruction complex (Figure 3B). Mutations in GSK3β phosphorylation sites of β-catenin, which are commonly observed in human cancers [119], renders the protein more resistant to p53-stimulated ubiquitination and subsequent proteasomal degradation [118].

Furthermore, p53 can also directly interact with GSK3β which modifies the structure and function of both interaction partners. In neuroblastoma cells, an association of GSK3β with p53 was observed upon chemotherapy-induced DNA damage [120], which led to an activation of GSK3β. Interestingly, this interaction occurs in a nuclear protein-complex, suggesting a function of GSK3β that is independent from its role in the cytosolic destruction complex. In another study, the C-terminus of the p53 basic domain and the N-terminus of GSK3β were found to be required for the mutual interaction, which resulted in a change of p53 acetylation [121]. The functional consequences of this p53-GSK3β interaction are complex and partly contradictory. For instance, the inhibition of GSK3β by either a dominant negative mutant or pharmacological inhibitors could attenuate p53-dependent induction of p21 and caspase-3 upon DNA damage [120]. This observation was confirmed by another study which showed that parallel treatment of colorectal cancer cells with the DNA damaging agent doxorubicin and GSK3β inhibitors could reduce p53-dependent p21 induction [122]. However, p53-dependent apoptosis was increased under these conditions, which relied on the BAX-associated mitochondrial apoptosis pathway. In contrast, another study observed that targeting GSK3β by RNAi or pharmacological inhibitors resulted in p53-dependent induction of p21 and apoptosis in colorectal cancer cells harboring wild-type p53, thereby inhibiting tumor growth in xenograft mice models [123].

Ablation of another component of the β-catenin destruction complex, CKIα, could trigger the activation of the p53 pathway. In a mouse model, this p53 activation counteracted the carcinogenic effects of Wnt hyperactivation, which was induced by the gut-specific ablation of *Csnk1a*, the mouse homologue of CKIα [124]. Additional ablation of p53 or its target gene p21 resulted in increased invasive tumor growth in this mouse model, indicating a network of p53 and Wnt signaling against intestinal carcinogenesis [124]. However, if CKIα and p53 also interact directly on a molecular level is yet unclear.

Furthermore, p53 was reported to promote the expression of the E3 ubiquitin ligase SIAH1 [125]. SIAH1 leads to the polyubiquitination of β-catenin through a multiprotein complex that also engages APC and thereby reduces Wnt activity [125,126,127] (Figure 3B). Through the SIAH1 axis, not only ectopically expressed p53, but also the endogenous p53, induced by UV irradiation or the DNA-damaging agent doxorubicin, could mediate β-catenin degradation [125]. In addition, hypoxia, a typical feature of the tumor microenvironment, was shown to reduce β-catenin levels by the upregulation of SIAH1 in a p53-dependent manner [128].

In summary, the interactions between p53 and the β-catenin destruction complex are highly diverse. The underlying mechanisms and the phenotypic consequences are not understood in all cases and may depend on the mutational background of the tumor models in which these interactions were observed.

### 5.2. Interactions of p53 with Secreted WNT Ligands and Antagonists

p53 was also found to interact with Wnt signaling via WNT ligands and secreted antagonists (Figure 3C). Webster et al. identified a positive feedback loop between p53 and non-canonical Wnt signaling in melanoma, showing that high WNT5A levels could stabilize wild-type p53 and increase its half-life, thus generating a slow-cycling, therapy-resistant cancer phenotype [129]. Furthermore, in lung cancer and glioma cells, an antagonizing network against tumor formation relying on p53 and DKK1, a secreted inhibitor of Wnt signaling, was described [130]. Wild-type, but not mutant p53, induced DKK1 expression upon DNA damage through binding to a responsive element located upstream of the DKK1 transcription start site [130] (Figure 3B). Recently, an interaction of p53 with canonical Wnt signaling via transcriptional control of WNT7B expression in hepatocellular carcinoma cells was described [131]. One subunit of TCP1, which is part of a chaperone complex, activates the expression of WNT7B and β-catenin by interacting with p53, thus influencing tumor proliferation and metastasis of hepatocellular carcinoma [131]. Moreover, it was reported that wild-type p53 could induce the transcription of WNT3, which in turn activates canonical Wnt signaling, and thereby promotes cancer stemness in different colorectal cancer models, including *Apc^Min/+/^Lgr5^EGFP^* mice and cancer organoids [132]. Similar activating effects of wild-type p53 on the expression of different secreted WNTs, particularly WNT3, were also observed in human and mouse embryonic stem cells [133,134]. These observations indicate that the transcriptional induction of secreted WNT ligands by p53 is not restricted to cancer but can be traced back to embryonic development.

### 5.3. Interactions of p53 with Wnt Transcription Factors

While manifold cross-talk of p53 with WNT ligands and destruction complex components were observed, much less is known about the interaction of Wnt transcription factors with p53. Rother et al. reported that TCF4, a major transcription factor of the Wnt/β-catenin pathway, is transcriptionally regulated by p53 [135]. Increased p53 levels were shown to downregulate the expression of TCF4 in cancer cells from different tissue backgrounds [135] (Figure 3B). This effect was dependent on wild type p53, and not observed in p53 mutants with deficient DNA binding sites. However, the underlying mechanism is not completely understood. No potential p53-binding sites were identified in the TCF4 promoter region, but p53 could downregulate the TCF/LEF-responsive reporter even in the presence of the degradation-resistant β-catenin mutant S33Y. This finding indicates that the downregulation of TCF4 by p53 is independent of changes in β-catenin levels. However, further studies are needed to further confirm this interaction between TCF4 and p53.

### 5.4. Mutant p53 Specific Interactions with Wnt Signaling

It has been long suggested that the mutational inactivation of TP53 could trigger the activation of Wnt signaling through promoting the aberrant accumulation of β-catenin [118,136,137]. However, not only loss-of-function, but also gain-of-function mutations of p53 with oncogenic properties were reported to interact with Wnt signaling. For instance, Kadosh et al. uncovered that the frequently occurring p53^R172H^ gain-of-function mutation could impose both tumor-suppressive and oncogenic effects in mouse models of Wnt-driven intestinal cancer (generated by *Csnk1a1* (encoding CKIα) deletion or *APC^min^* mutation) [138]. The p53^R172H^ mutant attenuated the formation of tumors in the proximal gut (duodenum and jejunum) while enhancing tumorigenesis in the ileum/colon. Mechanistically, p53^R172H^ interferes with the Wnt pathway by interrupting the TCF4-chromatin interaction, leading to tumor suppression and the promotion of differentiation in mouse tumor-derived organoids. Surprisingly, when the gut microbiome was eradicated by antibiotic treatment, p53^R172H^ exerted a tumor suppressive effect also in the distal gut, which was associated with reduced dysplasia and diminished Wnt signaling. The effect of the gut microbiome in counteracting p53 and promoting tumorigenesis was specifically mediated by a bacterial metabolite called gallic acid [138]. These findings demonstrate that environmental factors can influence the interaction between Wnt signaling and mutant p53 in cancer. In another mouse model of intestine carcinogenesis, the effect of the p53^R270H^ mutant, which is also considered as a gain-of-function mutation, was investigated. A combination of p53^R270H^ with Wnt activation by the Apc^Δ716^ background resulted in the development of tumors with accelerated submucosal invasion and enrichment of fibroblasts in the tumor stroma [94]. Interestingly, this phenotype was not observed when using p53 null mutants [94]. These mouse models show that gain-of-function mutations of p53 interact in a distinct manner with the Wnt pathway in colorectal cancer, leading to specific tumor phenotypes.

### 5.5. Interactions of p53 and Wnt Signaling Mediated by Non-Coding RNAs

MicroRNA (miRNA) and long non-coding RNAs (lncRNAs) are non-coding transcripts, sized respectively from 19–25 and >200 nucleotides [139,140], with important functions for tumor biology [141,142]. miRNAs are frequently downregulated in tumors and play an important role in tumor suppression [143], while lncRNAs act not only as suppressors, but also as oncogenes [144]. Several miRNAs/lncRNAs have been reported to participate in the cross-talk of p53 and Wnt pathway. For instance, miR-34, as a well-known direct transcriptional target of p53 [145], was linked to Wnt signaling in cancers [146]. Mechanistically, p53 suppresses canonical Wnt signaling through expression of miR-34, which targets several highly-conserved sites of untranslated regions in a set of Wnt pathway associated genes (*CTNNB1*, *WNT1*, *LRP6*, *WNT3* and *LEF1*), and leads to the repression of TCF/LEF activity in different cancer cell lines, including breast cancer, colorectal cancer and neuroblastoma [147,148]. Similarly, in colorectal cancer, miR-552 transduced hyperactive Wnt signaling to the transcriptional downregulation of p53 [149]. As an underlying mechanism, it was suggested that the β-catenin/TCF4 complex binds directly to the promoter region of miR-552 and thus enhances its expression [149]. For lncRNAs, it was recently reported that p53 could transcriptionally target the promoter of the lncRNA ST7 antisense RNA 1 (ST7-AS1) [150]. LncRNA ST7-AS1 inhibits Wnt target gene expression in human glioma cell lines, possibly through interaction with PTBP1, a splicing factor belonging to the subfamily of nuclear ribonucleoproteins. This inhibition of Wnt signaling induced by the p53-ST7-AS1 axis was able to repress proliferation, migration and invasion in glioma cells. These examples show that p53 and Wnt signaling are not only interacting through protein interactions but also through non-coding RNAs.

## 6. The Impact of p53-Wnt Cross-Talk on Cancer Phenotypes

In specific tumor entities, the interaction of p53 and Wnt signaling has a pivotal effect on different cancer phenotypes (Figure 4). The importance of this cross-talk for tumor initiation is highlighted by several genetically engineered mouse models. For instance, the adrenocortical tissue-specific expression of a gain-of-function β-catenin variant in combination with *Trp53* (the murine homologue of *TP53*) deletion resulted in the development of metastatic and hormonally active adrenocortical carcinoma, which shares similar gene expression profiles with primary human adrenocortical carcinoma [151]. This neoplastic transformation only occurred in mice harboring both alterations, emphasizing the synergistic effect of p53 and Wnt signaling for tumorigenesis. Similarly, the tissue-specific deletion of both *APC* and *TP53* resulted in the development of pancreatic mucinous cystic neoplasms [152], pancreatic acinar cell carcinoma [153], acute myeloid leukemia [154] and mammary neoplasms [155]. Notably, haploinsufficiency of *APC* was sufficient to drive carcinogenesis in some of these tumor models [153,154]. However, p53 inactivation can also abolish the dependence of tumor cells on external Wnt stimulation for proliferation. For instance, in a transgenic mouse model of WNT1-induced mammary adenocarcinoma, loss of one *Trp53* allele enabled the tumors to grow and progress independently of WNT1 stimulation [156]. Chromosomal instability (CIN) is common in many colorectal carcinomas [157]. In the established model of colorectal carcinogenesis, the adenoma-carcinoma-sequence, p53 puts a brake on cell proliferation and inhibits CIN after the initial *APC* mutation [83,158]. In a human colon organoid, CRISPR-based engineering of combinatorial *APC* and *TP53* loss resulted in the development of CIN [159].

Another link between Wnt signaling and p53 signaling was described for cancer stem cells, which are frequently implied in therapy resistance and metastasis, and often exhibit an upregulated Wnt pathway [68,160,161]. Physiological and aberrant p53 function have been shown to contribute to Wnt-driven cancer cell stemness. In colorectal cancer cell lines and tumor organoids, wild-type p53 stimulates the expression and secretion of WNT3 upon DNA damage by 5-fluorouracil, which drives cancer stemness and therapy resistance [132]. Likewise, a gain-of-function mutant of p53 was shown to induce cancer stemness in colorectal cancer by activating Wnt target genes in an *Apc^Δ716^ Trp53^R270H^* mouse model [94]. Furthermore, in colorectal cancer cell lines, BMP signaling can inhibit Wnt signaling, while the loss of *TP53* renders the cells resistant to BMP-mediated Wnt inhibition, possibly contributing to colorectal cancer progression [162].

The upregulation of components of the Wnt pathway by loss of p53 signaling has been shown to play an important role in metastasis formation. In genetically engineered mouse models of breast cancer, activation of Wnt signaling upon loss of *Trp53* could be shown. This Wnt activation was mediated by the increased expression and secretion of multiple WNT ligands, which stimulated tumor-associated macrophages and a systemic inflammatory environment that facilitated tumor metastasis. In line with this finding, inhibition of WNT secretion in this model could reduce pulmonary metastasis formation [163]. Not only secreted WNT ligands, but also the expression of their cognate receptors can be increased by TP53, thereby affecting cancer metastasis. In prostate cancer, deletion of *Trp53* has been shown to upregulate the WNT receptor FZD8, which in turn promotes invasion and migration in prostate cancer cell lines, but also bone metastasis in a prostate cancer mouse model [164,165]. These findings are further supported by an electroporation-based genetically engineered mouse model of prostate cancer [166]. Local overexpression of *Myc* and deletion of *Trp53* resulted in the formation of metastatic prostate cancers. Interestingly, the metastatic cancer cells acquired additional mutations in Wnt pathway components (*APC* mutations or amplifications of *LRP6* and *WNT2B*), underlining the importance of Wnt signaling for metastasis [166].

In several models of colorectal carcinogenesis, Wnt activation and loss of normal p53 function interact to promote tumor invasiveness. In a mouse model with intestine-specific *Apc^Δ716^ Trp53^R270H^* mutations, tumor invasiveness was found to be accelerated. Organoids derived from these invasive tumors showed an increased expression of Wnt genes such as WNT5B and FZD10 [94]. In another murine model of intestinal carcinogenesis, an induction of p53 signaling after Wnt hyperactivation by *Csnk1a1* ablation was observed. The additional deletion of *Trp53* drastically increased the invasiveness of the transformed cells, highlighting the interplay of Wnt hyperactivation and p53 abrogation for colorectal cancer cell invasion [124]. In summary, several cancer phenotypes are driven by the interaction of either wild-type or mutant p53 with Wnt/β-catenin signaling, highlighting the importance of both pathways and their synergistical action for cancer biology. In particular, metastasis and cancer invasiveness are likely facilitated by the Wnt/p53 cross-talk. The importance of the Wnt/p53 interaction is particularly relevant in colorectal cancer, as the sequential hyperactivation of Wnt signaling and the loss of p53 contribute to CIN as well as cancer stemness and tumor invasiveness.

## 7. Targeting the p53-Wnt Cross-Talk for Cancer Therapy

As both the p53 and Wnt pathways are critically involved in many cancer phenotypes, targeting their cross-talk has been considered as a promising anticancer approach. Efforts have been made, for instance, in colorectal cancer cell lines. In this tumor entity, Cheng et al. identified a novel inhibitor, labeled as compound 2, that could activate mitotic stress signaling, and lead to both the inhibition of canonical Wnt signaling and the activation of p53 [167]. In xenograft models, this compound could repress tumor growth with no obvious toxicity. Mechanistically, compound 2 was found to be a tubulin inhibitor, acting similarly to paclitaxel or vinblastine [167]. Recently, in hepatocellular carcinoma cells, the natural compound *trans*-chalcone was reported to increase p53 protein expression and decrease β-catenin levels, thereby inducing autophagic cell death and decreasing the metastatic capacity of HuH7.5 tumor cells [168]. Another study in glioblastoma cell lines showed that a histone deacetylase 8 inhibitor called NBM-BMX was able to downregulate Wnt/β-catenin signaling and to promote p53-mediated inhibition of the O6-methylguanine methyltransferase (MGMT) expression in glioblastoma cell lines [169]. This reduction of MGMT levels increased the sensitivity of tumor cells towards treatment with the alkylating chemotherapeutic agent temozolomide. Despite promising data, all these identified compounds have pleiotropic effects, and the specificity of this dual inhibition of the Wnt and p53 pathways, as well as the underlying molecular mechanisms, are not yet clarified.

Alternatively, combining chemotherapeutic agents that elicit a p53 response with inhibitors of the Wnt pathway is a potential strategy to target specific feedback loops. For instance, an increased efficacy was observed when 5-fluorouracil was combined with the WNT secretion/porcupine inhibitor LGK-974 for the treatment of colorectal cancer [132]. 5-fluorouracil is known to enhance p53 levels by increasing its translation and protein stability [170,171]. Cho et al. revealed that 5-fluorouracil treatment also increased Wnt signaling by stimulating the expression of WNT3, which caused the enrichment of cancer stem cells in colorectal cancers [132]. Interesting, this effect was dependent on p53, as the induction of cancer stem cells was not observed in isogenic colorectal cancer cells with loss of p53 or expression of the non-functional p53^R248W/−^ mutant. Furthermore, the biological effects caused by the 5-fluorouracil-induced WNT3 expression were inhibited by LGK-974. Concordantly, LGK-974 reduced regrowth of both patient-derived tumor organoids and cell lines after 5-fluorouracil treatment [132], indicating that targeting salvage pathways could be a potent approach for future combination therapies.

Pharmacological modulators of p53 stability were shown to have antineoplastic effects in preclinical cancer models [172]. One subgroup of p53 modulators, MDM2 inhibitors, stabilize wild-type p53 and induce cell cycle arrest and apoptosis [173]. MDM2 inhibitors are currently tested in advanced phase clinical trials as a treatment for hematological cancers [174]. Interestingly, they were also found to interact with Wnt pathway components in preclinical studies. For instance, the MDM2 inhibitor nutlin-3a could selectively reduce growth of *CTNNB1*-mutated adrenocortical cancer cells, but not of cells with wild-type *CTNNB1* [175]. In non-small cell lung cancer cells, nutlin-3 treatment was able to reduce TCF4 expression and thereby sensitize tumor cells to axitinib-induced apoptosis [176]. Furthermore, the beta-carboline-derivate SP141, a small molecule inhibitor of MDM2, could induce ubiquitination and degradation of β-catenin in pancreatic cancer cells, which was independent of the effect of the compound on MDM2 [177].

## 8. Outlook

Targeted cancer therapies aim at modulating core oncogenic and tumor-suppressive processes [178], but their clinical efficacy is often limited by a network of cellular feedback mechanisms. Deciphering the precise mechanisms of dysregulation and the associated interacting networks is, therefore, critical to improve cancer therapy. Both the p53 and Wnt pathways play key roles in cancer biology, as they have a broad impact on many cellular processes that are relevant for tumor survival and progression [179,180,181]. In this review, we summarized the current knowledge on the interaction of p53 and Wnt signaling in cancer. Both processes interact on multiple cellular levels, including direct protein interactions of core components, the regulation of protein stability and the transcriptional activation of key regulators. This manifold cross-talk can result in both positive and negative feedback loops, which translate to distinct cancer phenotypes. The quality of interaction, i.e., mutual activation or inhibition, is largely determined by two factors: the tissue background of the cancer model and the mutational status of Wnt pathway components and *TP53*. As highlighted in this review, specific alterations in *TP53* can either confer loss- or gain-of-function phenotypes. Similarly, genetic alterations of the WNT pathway can occur at either the ligand-receptor level or in genes of the destruction complex, resulting in differential effects on tumor biology. Accurate analysis of the combination of various *TP53* mutants and WNT pathway alterations in different tissue models is required to decipher the context-specific impact of the cross-talk on cancer phenotypes. A potential approach to disentangle the complexity of these genetic interactions is the combination of organoid culture with novel genome editing tools. In recent years, organoid cultures of many primary tissue types have been successfully established, including colon, liver and lung [182,183,184,185]. *De novo* introduction of mutations in these models could recreate key steps of carcinogenesis [186,187]. Using novel genome editing tools such as CRISPR base editing [188], it will be possible to rapidly introduce defined mutations in these normal tissue organoids, thereby creating isogenic models with different combinations of Wnt and p53 mutations. In-depth characterization of these models will enable a more profound insight into the biological consequences and underlying mechanisms of p53-Wnt cross-talk. As therapeutic strategies directed against both pathways have been largely unsuccessful in the past, understanding the tissue and mutational contexts of the interaction will help to develop more tailored and efficient treatment strategies against cancer.

## Figures and Tables

**Figure 1 biomolecules-12-00453-f001:**
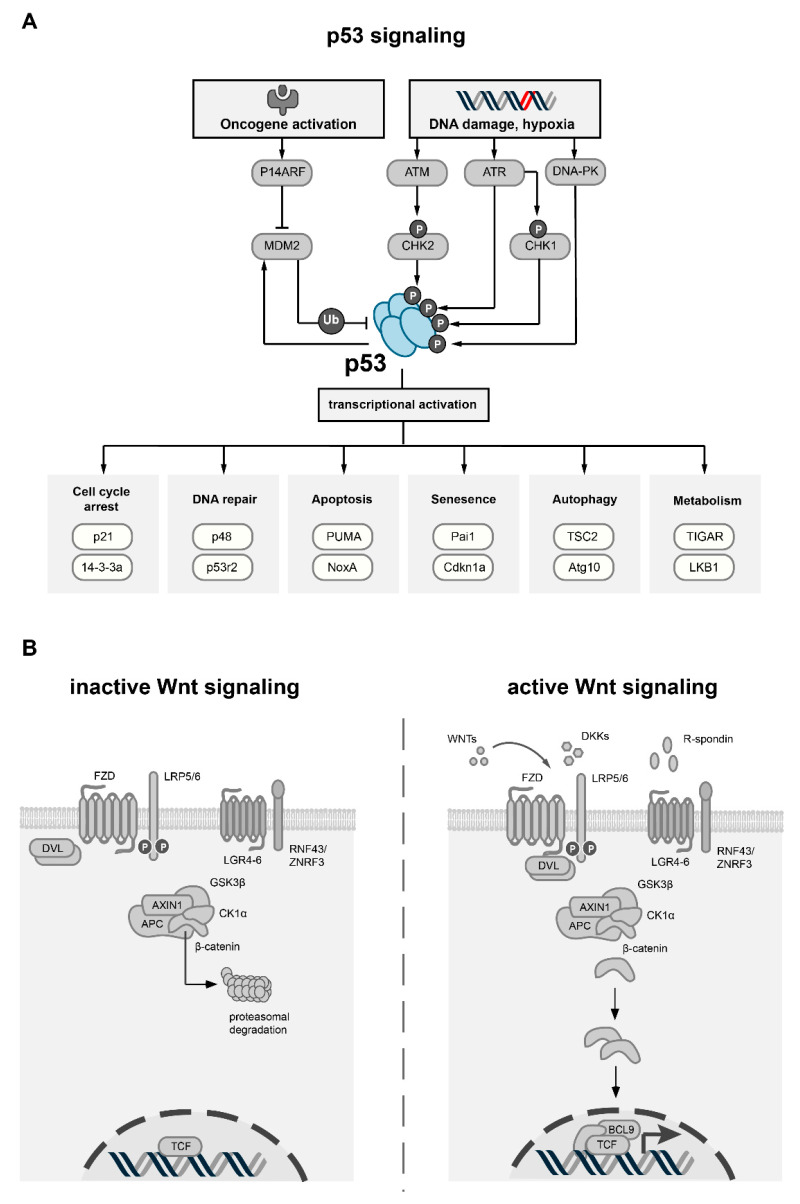
Overview of the p53 and Wnt pathway. (**A**) Schematic overview of p53 activation, regulation and transcriptional targets. Stress stimuli such as DNA damage or oncogene activation result in the activation of specific effector proteins (ATM, ATR, DNA-PK, CHK1/2, P14ARF), which stabilize p53 through phosphorylation or by inhibition of MDM2, the main negative regulator of p53. p53 itself can increase expression of MDM2, thereby creating a negative feedback loop. Activation of p53 increases the transcriptional activity of many target genes which are involved in key cellular processes (exemplary processes with target genes are shown). (**B**) Schematic overview of canonical Wnt signaling. In the inactive state (**left**), the absence of WNT ligands results in the phosphorylation of β-catenin by the destruction complex, which comprises the scaffold protein AXIN1, APC, GSK3β and CK1α. Upon phosphorylation by GSK3β, β-catenin is ubiquitinated and targeted for proteasomal degradation. Canonical Wnt signaling is activated upon binding of secreted WNT ligands to FZD receptors and LRP co-receptors (**right**). Plasma membrane levels of FZD and LRP receptors are regulated by secreted Wnt antagonist, such as DKK, or by the R-spondin/RNF43/ZNRF3 module. Upon binding of WNT ligands, LRP receptors are phosphorylated by CK1α and GSK3β, which leads to the activation of Dishevelled (DVL) proteins, thereby inactivating the destruction complex. As a consequence, β-catenin is stabilized, translocates to the nucleus and forms an active complex with LEF (lymphoid enhancer factor) and TCF (T-cell factor) transcription factors and co-activators such as BCL9, leading to the transcriptional activation of multiple target genes.

**Figure 2 biomolecules-12-00453-f002:**
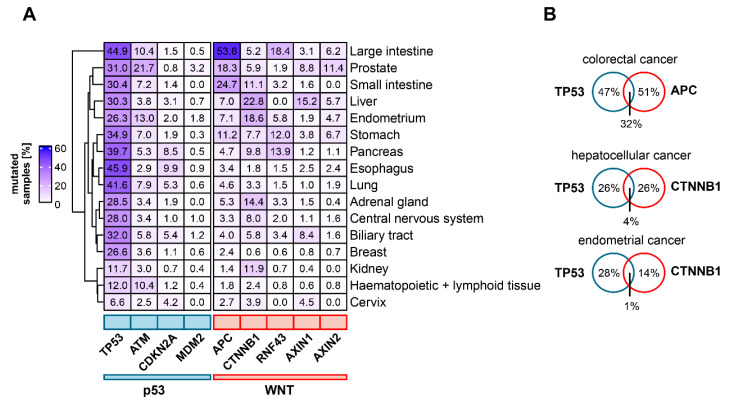
Mutational status of frequently mutated genes of the Wnt- and p53 pathways in different cancer entities. (**A**) Frequencies of mutations in Wnt and p53 pathway related genes in selected cancer entities. Percentage of samples with somatic mutations for the respective gene in the selected tissue type is shown. (**B**) Examples for mutual exclusive and co-occurring mutations in Wnt and p53 pathway associated genes. For Venn diagrams, only samples which were tested for mutations in both of the indicated genes were included. For this subset, the percentage of samples with mutations of either one or both genes were calculated. Tumor genome sequencing data were obtained from the COSMIC database [114].

**Figure 3 biomolecules-12-00453-f003:**
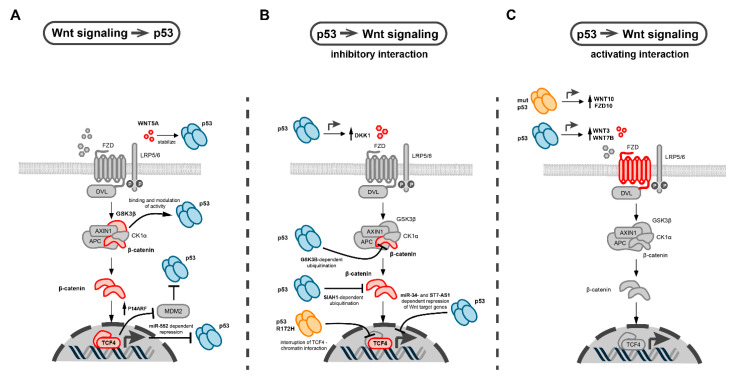
Molecular interactions between p53 and Wnt signaling in cancer. (**A**) Effects of Wnt pathway components on p53 function. These include β-catenin-dependent upregulation of P14ARF, an inhibitor of MDM2, the main negative regulator of p53. Other interactions include the modulation of p53 function by direct interaction with GSK3β, Wnt-dependent repression of p53 transcription via miR-52, and stabilization of p53 by WNT5A. Interacting Wnt pathway components are colored in red. (**B**,**C**) Biological effects of wild-type (blue symbol) and mutant p53 on different Wnt pathway components (colored in red). Inhibitory (**B**) and activating (**C**) interactions are shown. Arrows indicate activation and T-shaped arrow heads indicate inhibitory interactions. The main inhibitory effects of p53 on Wnt signaling include GSK3β and SIAH1-dependent ubiquitination of β-catenin, miRNA and lncRNA-mediated repression of Wnt target genes, and induction of secreted Wnt antagonists such as DKK1 (**B**). p53 stimulates Wnt signaling by increasing the expression of different WNT ligands and Fzd receptors (**C**).

**Figure 4 biomolecules-12-00453-f004:**
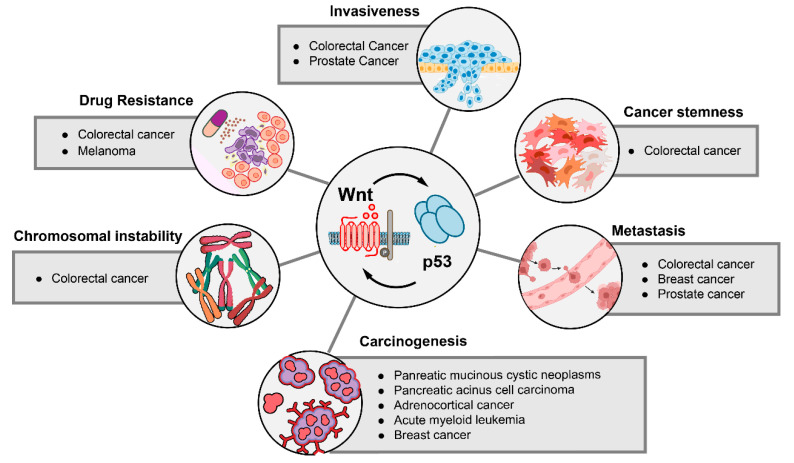
Cancer phenotypes that are regulated by the interaction of Wnt signaling and p53. Cancer phenotypes that depend on alterations of the Wnt and p53 pathways are shown. Tumor entities in which phenotypic effects of p53-Wnt cross-talk were detected are listed.

## Data Availability

Not applicable.

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
