# Peer review of "Cross-Talk between p53 and Wnt Signaling in Cancer"

_biomolecules, 2022, doi:10.3390/biom12030453_

Round 1

Reviewer 1 Report

Review

biomolecules-1624523

“Cross-talk between p53 and Wnt signaling in cancer”

Xiao et al.

Dear Authors,

Thank you very much for the oppertunity to be involved in the review process of the review article “Cross-talk between p53 and Wnt signaling in cancer”, by Xiao et al., submitted for publication in the Journal of Biomolecules.

In the following review article, the authors have provided a comprehensive overview of the multispectral molecular interactions between p53 and Wnt, the key regulatory molecules, orchestrating the cellular stress response system. Authors have described complex feedback loops and reciprocal transactivation of the cross-talk between p53 and Wnt pathways and how these correlates with different cancer phenotypes such as invasiveness, metastasis or drug resistance. Moreover, authors have summarized the mutational landscape of genes associated with p53 and Wnt signalling and discussed potential strategies to pharmacologically target the p53-Wnt interaction in order to improve cancer treatment.

In overall the review article is very detailed and comprehensive, with clear statements and logical connection. The current review article will be very relevant for the field as it provides enormous amount of information concerning both p53 and Wnt pathways and the cross-talk between components of both signalling networks. Therefore, I would like to suggest the article for publication in the journal. However, I would like to recommend a few improvements, which I believe will significantly increase readability of the review.

  1. It will be extremely helpful if authors provide a cartoon schematic, illustrating the normal activation (non-pathological) of the p53 and Wnt signalling pathways, including all the activities and proteins indicated in the manuscript. The high number of molecular factors with distinct functions and activities is quite difficult to follow if there is no adequate illustration of the entire signalling network and interactions between its components.

Moreover, it will be also beneficial if the figures (especially figure 2 and possibly the new figure) are additionally split into subsections (a, b, c), indicating distinct processes, in which p53 or Wnt components are involved. This will help when the figures are referred in the text at corresponding places. The reference to the corresponding figures should be more frequent, but not only once, in the beginning of the chapter. 

Author Response

We would like to thank the reviewer for carefully reading our manuscript and the positive comments.

It will be extremely helpful if authors provide a cartoon schematic, illustrating the normal activation (non-pathological) of the p53 and Wnt signalling pathways, including all the activities and proteins indicated in the manuscript. The high number of molecular factors with distinct functions and activities is quite difficult to follow if there is no adequate illustration of the entire signalling network and interactions between its components.

As suggested, we added a novel figure 1 to the revised version which visualizes the normal activation of the p53 and Wnt signaling pathway (page 4).

Moreover, it will be also beneficial if the figures (especially figure 2 and possibly the new figure) are additionally split into subsections (a, b, c), indicating distinct processes, in which p53 or Wnt components are involved. This will help when the figures are referred in the text at corresponding places. The reference to the corresponding figures should be more frequent, but not only once, in the beginning of the chapter. 

We subdivided the former figure 2 (now figure 3) into three separate subsections and added references to the respective subsections in the manuscript text (page 8). We hope that with these changes, the readability of the manuscript will be further improved.

Reviewer 2 Report

Dear Authors,

your review discussing the extensive cross-talk between p53 and Wnt signaling in cancer is well organized, thorough and clearly written. Nice schematics as well.

I have made a few suggestions (see pdf attached).

Regards

Author Response

We would like to thank the reviewers for the positive comments on our manuscript. The suggested references were added to the revised version. Furthermore, we adapted the discussion section to make it more concise, but also included novel aspects that were demanded by another reviewer.

Reviewer 3 Report

In the manuscript “Cross-talk between p53 and Wnt signaling in cancer” the authors propose to summarize the functional consequences of p53-Wnt for cancer phenotypes such as invasiveness, metastasis or drug resistance and discuss potential strategies to pharmacologically target the p53-Wnt interaction. The paper is of great interest and the text of good quality. I have only minor suggestions.

Specific comments:

  1. Overall the text is of excellent quality but in some of the parts I would recommend the authors to specifically refer to works that show the conclusions discussed. It would be great to include some inputs on the experimental design/approach and discuss future perspectives.
  2. When discussing mutational landscape of p53 and Wnt pathway genes in cancer it would be interesting to include some information regarding the aggressiveness associated with that landscape.
  3. p53 modulators could be briefly analysed and discussed. It would add great interest to the text.

Author Response

We would like to thank the reviewer for the valuable suggestions.

Overall the text is of excellent quality but in some of the parts I would recommend the authors to specifically refer to works that show the conclusions discussed.

We carefully went through the manuscript and added several additional references to support our statements, especially in the general part on p53 and Wnt signaling (e.g. page 3, line 109, 112, 123).

It would be great to include some inputs on the experimental design/approach and discuss future perspectives.

In the revised discussion section, we added potential future strategies to analyse the interaction of Wnt-p53 using experimental methods such as organoid culture and CRISPR technology (page 13). 

When discussing mutational landscape of p53 and Wnt pathway genes in cancer it would be interesting to include some information regarding the aggressiveness associated with that landscape.

We now added a concise subsection on the impact of p53 and Wnt pathways mutations on aggressiveness and survival of cancer patients (page 6, line 225-243).

p53 modulators could be briefly analysed and discussed. It would add great interest to the text.

In the revised manuscript, a subsection was added that describes the impact of p53 modulators, specifically MDM2 inhibitors, on the p53-Wnt crosstalk in cancer (page 13, line 519-530).